# Physical Therapy Utilization and Morbidity Outcomes After Breast Cancer Surgery: A Longitudinal Analysis of Three Combined Cohorts

**DOI:** 10.3390/cancers17203296

**Published:** 2025-10-11

**Authors:** Ifat Klein, Danit R. Shahar, Michael Friger, Irena Rosenberg, Daphna Barsuk, Merav A. Ben-David, Sergio Susmallian

**Affiliations:** 1Physical Therapy Department, Assuta Medical Centers, Tel Aviv 6971028, Israel; irena@assuta.co.il; 2Department of Epidemiology, Biostatistics and Community Health, School of Public Health, Faculty of Health Sciences, Ben-Gurion University of the Negev, Beer-Sheva 8410501, Israel; dshahar@bgu.ac.il (D.R.S.); friger@bgu.ac.il (M.F.); 3Breast Surgery Department, Assuta Medical Centers, Tel Aviv 6971028, Israel; barsok_da@mac.org.il; 4Faculty of Health Sciences, Ben-Gurion University of the Negev, Beer sheva 8410501, Israel; meravak@assuta.co.il; 5Oncology Department, Assuta Medical Centers, Tel Aviv 6971028, Israel; 6Surgery Department, Assuta Medical Centers, Tel Aviv 6971028, Israel; sergios@assuta.co.il

**Keywords:** breast cancer surgery, upper extremity morbidity, physical therapy utilization, rehabilitation access, timely intervention

## Abstract

Many women develop arm and shoulder problems after breast cancer surgery, such as pain, lymphedema, or reduced mobility. These complications often appear gradually and can limit daily life. In our study of 1602 women across multiple centers, we followed recovery for up to three years after surgery. We found that only 48% received physical therapy, and almost 60% of them started treatment later than three months after surgery. Women who had more extensive surgery or developed lymphedema were more likely to use physical therapy, but even then, referrals often came too late. This shows that physical therapy is both underused and delayed, despite being effective in reducing disability. A structured system of risk-based referrals and scheduled rehabilitation follow-ups could ensure earlier access to care and help survivors recover better.

## 1. Introduction

Physical recovery from breast cancer frequently requires addressing the functional consequences and treatment-related side effects resulting from both surgical interventions and systemic oncological therapies [1]. Unlike the immediate postoperative phase, which is typically well-defined and medically monitored, the long-term recovery trajectory is more variable and often underestimated [2]. Many complications do not present directly after surgery but rather develop insidiously over weeks or even months, continuing well into survivorship [3]. This pattern of delayed morbidity poses a unique challenge to both patients and clinicians [4].

Upper extremity morbidities are among the most common and impactful consequences of breast cancer treatment [5]. Surgical interventions, particularly axillary lymph node dissection, remain central to oncologic control but come at a cost—often resulting in pain, reduced shoulder range of motion, and functional limitations [6]. When combined with adjuvant therapies such as chemotherapy, radiation, and targeted biologic treatments, the risk of musculoskeletal and soft tissue complications increases further [7]. One of the most prevalent and burdensome late-onset conditions is lymphedema—a chronic, progressive swelling of the arm that can impair function, limit daily activities, and significantly reduce quality of life [8]. While improvements in surgical techniques and radiotherapy planning have led to a decline in lymphedema incidence, its gradual and often irreversible nature underscores the need for proactive management [9].

Another frequently overlooked complication is axillary web syndrome, also known as lymphatic cording. Axillary web syndrome typically develops within the first few weeks after surgery and is characterized by the appearance of tight, cord-like structures extending from the axilla into the arm. Though the condition often resolves spontaneously within several months, it can cause significant pain, limited shoulder mobility, and functional distress during its course [10]. Physical therapy has been shown to effectively alleviate axillary web syndrome symptoms and improve shoulder range of motion [11], while the quality and implementation of structured rehabilitation programs tailored to oncologic recovery can vary significantly across institutions, potentially influencing recovery timelines.

Despite accumulating evidence demonstrating the benefits of early physical therapy [12,13,14], most patients begin physical therapy only after the development of significant morbidity—such as persistent pain, severe functional limitation, or diagnosis of lymphedema or axillary web syndrome [15,16]. This delay may reduce the effectiveness of rehabilitation and prolong the duration of disability—both in terms of symptom mitigation and long-term functional outcomes. 

In this context, early identification of high-risk patients, combined with timely and individualized rehabilitation strategies, is essential [13]. Screening tools and clinical predictors can guide targeted interventions, enabling healthcare providers to shift from reactive to proactive care [17]. By optimizing the timing and type of rehabilitation offered, it may be possible to enhance functional outcomes, reduce the burden of long-term complications, and support a more efficient return to daily activities and overall well-being.

## 2. Objective

This study aims to explore the timeline of morbidity development following breast cancer surgery, characterize the timing of patients’ initiation of physical therapy, and identify clinical and contextual factors associated with the utilization of rehabilitation services.

## 3. Methods

A retrospective study was conducted at five Assuta medical centers from October 2022 to June 2024. Data were pooled from two studies that evaluated postoperative morbidity at different time periods and were approved by the Assuta Medical Centers Ethics Committee (-0019-22 and 0018-23). A subset of participants underwent prospective follow-up.

### 3.1. Participants

Women aged 18–80 with a first primary breast cancer diagnosis who underwent elective oncologic surgery 0–36 months prior were eligible. Exclusions: metastatic disease, prior non-breast malignancy, benign breast tumors, severe cognitive impairment, or bedridden status.

### 3.2. Data Collection and Measures

After providing electronic informed consent, participants completed a smartphone-based survey and we abstracted clinical data from the electronic records.

### 3.3. Arm Morbidity

QuickDASH. The 11-item Quick Disabilities of the Arm, Shoulder and Hand (QuickDASH) was scored 0–100, with higher scores indicating worse upper-limb disability. For clinical interpretation we considered a minimal clinically important difference of ~16 points (anchor-based; exceeds the minimal detectable change of ~13). In regression models, QuickDASH was analyzed per 10-point increment [18].

Pain. Numeric Pain Rating Scale, 0–10 over the past week [19].

Range of motion limitation. Self-reported limitation in shoulder or arm range of motion (yes/no).

Lymphedema. Clinician-diagnosed (surgeon/oncologist/physiotherapist) upper-limb lymphedema recorded in the medical record (yes/no).

Functional decline. Functional decline was defined a priori as QuickDASH > 20, a conservative threshold that exceeds the instrument’s minimal clinically important difference of ≈16 points [20].

### 3.4. Risk Screening for Future Arm Morbidity

The Arm Morbidity following Breast Cancer Treatments (ARM-BCT) is a 17-item screening instrument that integrates patient- and treatment-related factors into a 1–20 risk score (higher = greater risk). Scores are classified as low (<6), intermediate (6–7), or high (>7) risk, enabling risk-stratified recommendations on the need and timing of rehabilitation (Appendix A) [18].

### 3.5. Emotional Well-Being

Selected SF-36 subscales were administered and transformed to standard 0–100 scores, where higher values indicate better health [21]. Anxiety symptoms were each rated 0–10 on a numeric scale (higher = worse). For analyses, anxiety was modeled per 1-point increase (0–10).

Physical activity. Physical activity was assessed using a brief questionnaire inspired by the Godin–Shephard Leisure-Time Exercise Questionnaire [22]. Participants reported weekly frequency (sessions/week) of light, moderate, and vigorous activity in three items. We derived a study-specific composite: (light × 1 + moderate × 2 + vigorous × 3), reported as weighted sessions/week (higher = more activity).

Clinical and treatment variables. Age (years), body-mass index (BMI, kg/m^2^), comorbidity (neurological or orthopedic problems that limit function, prior surgery or injury to the shoulder on the operated side, diagnosed with fibromyalgia, chronic pain syndrome (yes/no), surgery type (lumpectomy, mastectomy, prophylactic mastectomy), axillary surgery extent and chemotherapy, radiotherapy (yes/no).

Time since surgery. Postoperative interval at survey was categorized as 0–6, 7–12, 13–24, and 25–36 months; 0–6 months served as the reference in regression unless stated otherwise.

### 3.6. Statistical Analysis

Cohort characteristics and PROs by postoperative interval (0–6, 7–12, 13–24, 25–36 months) were compared using ANOVA or Kruskal–Wallis and χ^2^ tests. Univariate logistic models assessed factors linked to any physical therapy uptake. Multivariable logistic regression identified independent predictors of physical therapy utilization, adjusting for demographics, comorbidity, surgery type, and symptom scores. We then examined the effect of early (≤3 months) versus late (>3 months) physical therapy initiation on recovery outcomes in adjusted models. For interpretability, continuous predictors were reported per a clinically meaningful unit: QuickDASH per 10 points; pain and anxiety per 1 point; Arm Morbidity following Breast Cancer Treatments (ARM-BCT) per 1 point. Odds ratios (OR) and 95% confidence interval (CIs) are presented accordingly. Significance was set at *p* < 0.05; analyses were performed in SPSS 29.

### 3.7. Reporting Guideline

This study follows the STROBE (Strengthening the Reporting of Observational Studies in Epidemiology) reporting guidelines. The completed STROBE Checklist is provided as Appendix A.

## 4. Results

A total of 6106 women were approached for participation, of whom 1876 (30.7%) agreed and completed the study questionnaire. Among these, 274 were excluded: 44 due to metastatic disease, 42 because of additional surgery or injury affecting arm function, 6 for a pre-existing diagnosis of lymphedema, and 182 owing to non-malignant breast conditions (e.g., fibroadenoma). The final analytic cohort comprised 1602 patients (Figure 1).

The cohort (n = 1602) had a mean age of 57.1 ± 12.2 years (median 58, range 18–79) and a mean BMI of 26.4 ± 5.3 kg/m^2^ (median 25, range 14.0–69.0; n = 1568).

Most women underwent lumpectomy (76.9%), with 19.6% receiving mastectomy and 3.5% prophylactic mastectomy. Sentinel-node biopsy (1–4 nodes) was performed in 76.8%, axillary dissection (≥5 nodes) in 14.8%, and no node removal in 8.4%. Breast reconstruction was carried out in 15.3% of cases. Full cohort characteristics are detailed in Table 1.

In this cohort of 1602 women, the median age was 58 years interquartile range (IQR 50–67) and median BMI was 25 (IQR 22–29). Most underwent breast-conserving surgery (76.9%), 19.6% had mastectomy, and 3.5% underwent prophylactic mastectomy. Sentinel/limited nodal sampling (1–4 nodes) was performed in 76.8%, axillary dissection (≥5 nodes) in 14.8%, and 15.3% received breast reconstruction. The ARM-BCT score had a median of 6 (IQR 4–8). QuickDASH was 13 (IQR 0–17) (n = 1184 available cases).

Analysis of symptom and function-related variables by time since surgery (Table 2) revealed a gradual decline in morbidity over time, though many symptoms persisted well beyond the acute phase. Anxiety levels were highest shortly after surgery and declined steadily over time (*p* < 0.001).

Figure 2 models timing of postoperative survey follow-up (appearance in later vs. 0–6-month windows). Forest plots show adjusted odds ratios (ORs) with 95% confidence intervals (CIs) from a multinomial logistic model of follow-up timing, stratified by postoperative intervals—(A) 7–12, (B) 13–24, and (C) 25–36 months—relative to 0–6 months (reference). Points denote ORs; horizontal bars denote 95% CIs. Values > 1 (to the right of the dashed vertical line at OR = 1) indicate higher odds of follow-up in that interval versus the reference; values < 1 indicate lower odds. The x-axis is log-scaled. Abbreviations: OR, odds ratio; CI, confidence interval; ROM, range of motion; AWS, axillary web syndrome.

In multinomial regression modeling (Figure 2), delayed follow-up (13–24 and 25–36 months versus 0–6 months) was strongly associated with higher pain scores: each one-point increase in pain raised the odds of appearing in those later intervals (*p* < 0.0001). Lymphedema was associated with increased odds of follow-up in the 25–36-month window (OR ≈ 4.3; *p* = 0.027). In contrast, elevated anxiety and the presence of axillary web syndrome were linked to significantly lower odds of follow-up beyond the first 6 months after surgery. Anxiety reduced the likelihood of delayed follow-up (p < 0.001), and axillary web syndrome markedly decreased the odds of being seen in both the 13–24 and 25–36-month intervals (both *p* < 0.001). This likely reflects the early onset of axillary web syndrome —often within the first few weeks after surgery—which typically prompts earlier clinical attention and rehabilitation referral.

When comparing women who did and did not receive postoperative physical therapy (Table 3) recipients were more likely to have undergone lumpectomy (81.6% vs. 71.6%) and less likely to have had mastectomy or prophylactic mastectomy (overall *p* = 0.008), with no significant differences in nodal procedure (*p* = 0.140) or breast reconstruction (*p* = 0.254). They reported higher risk and disability (ARM-BCT: median 6 [IQR 4–8] vs. 5 [3–8]; QuickDASH: 15 [12–18] vs. 11 [0–16]; both *p* < 0.001) and more comorbidity (45.6% vs. 31.6%, *p* < 0.001). Range of motion limitation was similar between groups (36.4% vs. 36.1%, *p* = 0.478), whereas decreased function was more frequent among non-recipients (44.4% vs. 36.8%, *p* = 0.015). Lymphedema did not differ significantly (9.7% vs. 7.1%, *p* = 0.136), while axillary web syndrome was more common among non-recipients (14.2% vs. 8.2%, *p* = 0.002). Pain levels were comparable (median 1 in both; *p* = 0.820), and anxiety was lower among physical therapy recipients (median 1 [1,2,3] vs. 2 [1,2,3,4,5], *p* < 0.001). Inpatient physical therapy instruction did not differ significantly (50.9% vs. 45.5%, *p* = 0.086), whereas a verbal recommendation from medical staff was strongly associated with receiving outpatient physical therapy (28.2% vs. 8.3%, *p* < 0.001). Age and BMI distributions were similar between groups (both *p* > 0.10).

Figure 3 Adjusted predictors of physical therapy utilization. Forest plot of adjusted odds ratios (ORs) with 95% confidence intervals (CIs) from the multivariable model. Continuous predictors are reported as QuickDASH per 10 points and pain per 1 point. The x-axis is on a log scale; the dashed red line marks OR = 1. All predictors shown were statistically significant (95% CIs did not cross 1).

In the multivariable analysis (Figure 3; full estimates in Appendix A), longer intervals since surgery were independently associated with lower odds of initiating outpatient physical therapy (adjusted OR per 6-month increment, 0.76; *p* = 0.003). In contrast, more extensive surgery—mastectomy (OR 2.10; *p* = 0.028) and axillary lymph-node dissection (OR 2.64; *p* = 0.001)—and the development of lymphedema (OR 3.06; *p* = 0.004) were strongly associated with increased physical therapy uptake. Receipt of in-hospital physical therapy (OR 1.36; *p* = 0.048) and engagement in complementary therapies (OR 2.82; *p* < 0.001) further predicted greater utilization. Higher pain scores (OR 0.87 per unit; *p* = 0.007) were paradoxically associated with reduced odds of follow-up therapy. Other factors, including comorbidity, breast reconstruction, functional measures, QuickDASH scores, and range of motion limitation, showed no independent effect. Overall, these findings highlight a critical early postoperative “window of opportunity” for initiating rehabilitation and underscore surgical and symptom-related factors that may warrant proactive referral for physical therapy.

Table 4 presents the clinical factors independently associated with physical therapy initiation timing after breast cancer surgery. In the multinomial model (complete cases n = 1114; reference outcome: Early physical therapy ≤ 3 months), higher pain scores independently increased the odds of both no physical therapy (adjusted OR = 1.11 per 1-point increase, 95% CI 1.01–1.22, *p* = 0.037) and late physical therapy (>3 months vs. early; OR = 1.21, 95% CI 1.11–1.33, *p* < 0.001). Lymphedema was likewise associated with higher odds of no physical therapy (OR = 2.59, 1.10–6.07, *p* = 0.029) and late physical therapy (OR = 3.04, 1.31–7.04, *p* = 0.010). Axillary web syndrome did not relate to omission (OR = 0.87, *p* = 0.587) but predicted late physical therapy (OR = 2.48, 1.42–4.34, *p* = 0.001). Other variables—including range of motion limitation, decreased function, anxiety, physical activity, QuickDASH, and ARM-BCT—were not significantly associated with physical therapy timing; ARM-BCT showed a borderline association for no physical therapy (OR = 1.08 per point, 1.00–1.18, *p* = 0.062).

## 5. Discussion

This study delineates the temporal course of upper-extremity morbidity following breast cancer surgery and examines how physical therapy utilization aligns—or fails to align—with evolving clinical needs. The findings underscore that recovery in this population is neither linear nor confined to the immediate postoperative period. Rather, conditions such as lymphedema, axillary web syndrome, persistent pain, and functional impairments often arise weeks to months after surgery and may persist for years, distinguishing breast cancer recovery from that of other surgical populations.

Physical therapy has been shown to play a beneficial role in the recovery of women following breast cancer surgery, particularly in addressing common treatment-related complications [23]. Evidence from previous studies indicates that physical therapy interventions can effectively reduce pain, improve shoulder range of motion, restore functional capacity, and enhance muscular strength [24]. These benefits are especially critical in managing sequelae such as post-surgical stiffness, axillary web syndrome, and lymphedema, which may otherwise persist and impair long-term quality of life [14]. Moreover, early integration of physical therapy into postoperative care has been associated with faster recovery trajectories and better reintegration into daily activities [25,26]. Despite these established advantages, access to timely rehabilitation remains inconsistent, highlighting a need for structured referral pathways and increased awareness of its therapeutic value [27,28,29].

The morbidity trajectory observed in our cohort of 1602 women highlights a prolonged and dynamic symptom profile. Although axillary web syndrome and lymphedema were most frequently reported within the first six months, new or persistent cases were observed up to two years postoperatively. Even in the 25–36-month window, approximately one-third of participants continued to report range of motion limitations or decreased upper-limb function—emphasizing the chronicity of these complications. These results are consistent with earlier studies. For instance, Johnson et al. documented persistent impairments in 25–35% of patients up to two years after treatment, while Lee and Carpena-Niño also noted sustained functional decline. De Groef et al. further reported that axillary web syndrome may initially present within three months but can persist or reappear beyond one year—patterns mirrored in our data [30]. Similarly, lymphedema developed months after surgery in approximately 20% of patients undergoing axillary dissection, consistent with prior epidemiologic estimates [31]. This reinforces recommendations, such as those by Brunelle, to implement prospective screening programs that enable early detection and intervention [9].

Comorbidities were significantly more common among women who did not receive physical therapy (31.6% vs. 45.6%). This counterintuitive finding suggests that patients with multiple health conditions may face greater barriers to rehabilitation, including competing medical priorities, treatment fatigue, and logistical constraints. Although these women may have a heightened clinical need, comorbidities may ultimately limit both referral and engagement in rehabilitation programs. Similar patterns were described by Abbad-Gomez et al., who reported that higher comorbidity burden reduced adherence to follow-up and rehabilitation services [32]. These results highlight the importance of tailoring referral strategies and supportive care pathways for patients with complex medical profiles to ensure equitable access to physical therapy.

A paradoxical association emerged whereby higher pain scores were linked to a lower likelihood of physical therapy utilization. This counterintuitive finding may reflect that women with substantial pain are more often managed with pharmacologic strategies rather than referred for rehabilitation, or that pain itself restricts mobility and thereby limits access to outpatient services. Similar patterns have been noted in survivorship research, where pain severity was associated with reduced participation in nonpharmacologic interventions [33]. These observations underscore the need for clinicians to recognize pain not only as a symptom to be alleviated but also as a potential barrier to rehabilitation engagement. Embedding pain assessment into structured referral pathways may help ensure that patients with greater symptom burden are proactively directed toward physical therapy rather than excluded from it.

Another key finding was that nearly 60% of women who eventually engaged in physical therapy began treatment more than three months after surgery. Delayed initiation may substantially reduce the effectiveness of rehabilitation, as prior trials have shown that early intervention is critical for preventing persistent morbidity and optimizing functional recovery [13]. Women with lymphedema, axillary web syndrome, and higher pain scores were particularly likely to initiate physical therapy late rather than early, indicating that referrals were often reactive to symptom escalation rather than proactive [34]. This pattern reflects the absence of systematic follow-up and highlights the need for structured, risk-based pathways that ensure rehabilitation is initiated within the critical early postoperative window, before complications become entrenched.

Taken together, these findings demonstrate that neither symptom burden nor clinical risk alone reliably drive physical therapy use. Instead, utilization patterns appear largely reactive, occurring only once morbidity becomes clinically apparent. This systemic underutilization aligns with international reports, where physical therapy uptake among breast cancer survivors remains inconsistent and frequently delayed. Cheville et al. (USA) found that only 35% of women with upper-limb morbidity were referred for rehabilitation [35], while Keesing et al. (Australia) [36] and Lin and Schmitz reported similarly fragmented access [37,38]. These parallels highlight that the gaps observed in our cohort reflect a broader, global challenge in survivorship care.

Clinical implications: The findings underscore an urgent need to embed rehabilitation into structured oncology follow-up rather than leaving it dependent on patient initiative or symptom escalation. Risk-stratified referral protocols, ideally supported by electronic health record prompts, could ensure that women with high-risk surgical profiles or early morbidity indicators are referred proactively.

A compelling example is the prospective surveillance model, developed in leading U.S. institutions [39], which incorporates preoperative baseline assessments and structured follow-ups at 1, 3, 6, and 12 months post-surgery [40]. Studies by Stout et al. and Fu et al. demonstrated that such early, structured intervention reduces the incidence of clinical lymphedema, improves function, and lowers long-term healthcare costs [41,42]. Adoption of similar systems—potentially supported by tele-rehabilitation platforms and EHR-embedded clinical decision aids—could help close existing care gaps and ensure timely, equitable access.

Patient-level barriers may further limit engagement, as shown in prior qualitative studies that identified time constraints, transportation challenges, caregiving responsibilities, and, in some cases, lack of interest [43]. Addressing both system-level and individual-level barriers will be essential to maximize the impact of rehabilitation programs.

Our study also demonstrated that even among high-risk subgroups—such as those undergoing mastectomy, axillary dissection, or reconstruction—physical therapy was not consistently initiated early. Although these surgical characteristics were independently associated with greater morbidity and an increased likelihood of receiving therapy, the timing of intervention was frequently suboptimal. Notably, women with elevated risk profiles—reflected in higher ARM-BCT scores—did not receive rehabilitation earlier than others. This disconnects between predicted morbidity and the actual delivery of care underscores the need for structured, risk-based referral protocols to support timely intervention.

Together, these findings emphasize the need to reframe rehabilitation as a core component of survivorship care, delivered proactively and equitably to women at risk of long-term morbidity.

## 6. Strengths and Limitations

This multicenter study leverages a large cohort and a time-segmented design (0–6, 7–12, 13–24, 25–36 months), integrating clinical variables with patient-reported outcomes and prespecified multinomial models with transparent unit interpretation. Limitations include the retrospective design and available-case analyses with window-dependent denominators, which may introduce selection and missing-data biases. Several outcomes are self-reported (e.g., pain, anxiety, range of motion limitation, axillary web syndrome, lymphedema), and we lacked standardized lymphedema staging/severity or objective arm-volume measures. The cohort—predominantly younger, early-stage patients from a single health system—may limit generalizability. Our physical activity measure is inspired by the Godin–Shephard Questionnaire—weighted sessions per week; 1/2/3 scheme rather than the instrument’s validated score; accordingly, this adaptation was not validated and lacks external cut-points which limits cross-study comparability. We did not capture system-level factors (access, provider availability, financial barriers) nor physical therapy content, dose, or adherence. Given the retrospective design, causality cannot be inferred; taken together, these factors likely bias toward underestimation of late morbidity and under-referral, reinforcing the need for early assessment and planned follow-up.

## 7. Conclusions

Morbidity frequently emerged or persisted months after surgery, yet physical therapy was under-utilized and often initiated late. Higher pain and lymphedema were linked to delayed or omitted physical therapy, and axillary web syndrome to late physical therapy, underscoring the need for proactive, risk-stratified referral and structured follow-up within the first 3 months to avoid reactive care. Incorporating tools such as the ARM-BCT and electronic health record prompts may facilitate timely referral and reduce long-term disability. Health systems should prioritize early assessment and planned follow-up as core health-promotion policy elements within survivorship care to reduce preventable disability.

## Figures and Tables

**Figure 1 cancers-17-03296-f001:**
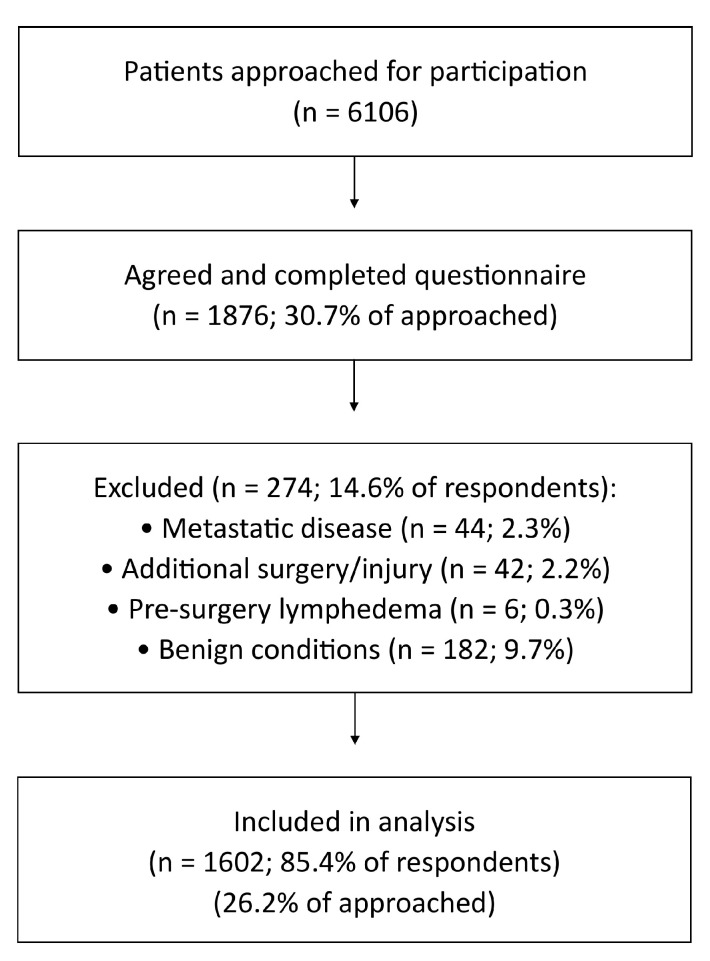
Recruitment and Inclusion Flowchart of Study Participants. Study flow diagram. Flow of patient inclusion and exclusion. Boxes report counts (n) and percentages at each stage. Total excluded: 274.

**Figure 2 cancers-17-03296-f002:**
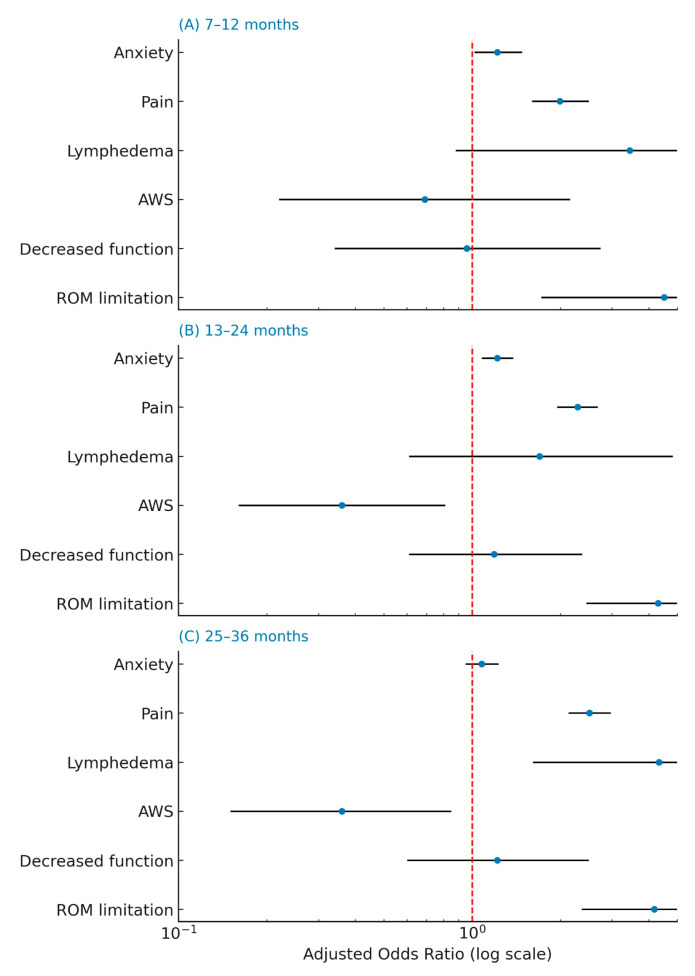
Clinical predictors of timing of postoperative follow-up.

**Figure 3 cancers-17-03296-f003:**
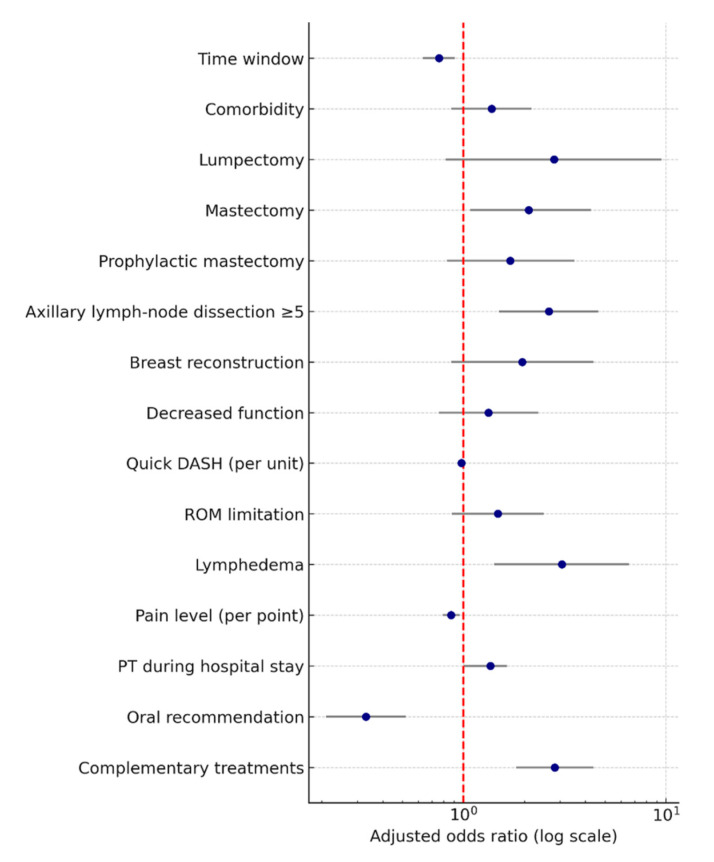
Clinical predictors of postoperative physical therapy utilization. Abbreviations. OR, odds ratio; CI, confidence interval; PT, physical therapy; ROM, range of motion; QuickDASH, Quick Disabilities of the Arm, Shoulder and Hand questionnaire.

**Table 1 cancers-17-03296-t001:** Sociodemographic, Clinical, and Treatment Characteristics of the Study Cohort.

Sociodemographic Characteristics			
Age (n = 1602)	median 58 IQR [50;67]		
BMI (n = 1568)	median 25 [22;29]		
Family status (n, %)	Single	358	23.4
	Married	897	58.7
	Divorced	168	11.0
	Widowed	106	6.9
Years of education (n, %)	8–12	295	20.1
	13–16	551	37.6
	17–18	363	24.7
	19<	258	17.6
BRCA gene, positive (n, %)		254	19.9
Smoking, yes (n, %)		229	15.7
Clinical Characteristics			
Surgery type (n, %)	Lumpectomy	1091	76.9
	Mastectomy	279	19.6
	Prophylactic mastectomy	50	3.5
Lymph node dissection (n, %)	No nodes removed	102	8.4
	1–4	931	76.8
	≥5 nodes	180	14.8
Breast reconstruction, yes (n, %)		211	15.3
Stage (n, %)	I	982	88.4
	II	85	7.7
	III	42	3.8
Comorbidity, yes (n, %)		614	38.1
ARM-BCT score (n = 1571)	median 6 IQR [4;8]		
QuickDASH (n = 1184)	median 13 IQR [0;17]		
Physical activity score (n = 1062)	median 2 IQR [2;8]		
Treatment Characteristics			
Chemotherapy, yes (n, %)		302	20.5
Radiation therapy, yes (n, %)		525	35.7

Abbreviations: BMI, body mass index; IQR, interquartile range; BRCA, pathogenic BRCA1/2 variant; ARM-BCT, Arm Morbidity after Breast Cancer Treatments (higher = greater risk); QuickDASH, Disabilities of the Arm, Shoulder and Hand (0–100; higher = greater disability). Note: Continuous variables are summarized as median [IQR] using available cases; categorical variables as n (%). Percentages use non-missing denominators for each variable. Physical Activity Score (GLTEQ-inspired): weighted sessions/week = (light × 1 + moderate × 2 + vigorous × 3); higher = more activity.

**Table 2 cancers-17-03296-t002:** Morbidity symptoms reported by time elapsed since breast cancer surgery.

	Time Range (Months)				*p* Value
	0–6 N = 419	7–12 N = 80	13–24 N = 733	25–36 N = 370	
ROM limitation, yes n (%)	101 (31.7)	24 (45.3)	116 (40.3)	87 (35.7)	0.145
Decreased function, yes n (%)	124 (38.8)	26 (49.1)	120 (41.7)	84 (34.6)	0.273
Lymphedema, yes n (%)	25 (7.8)	3 (5.7)	21 (7.3)	23 (9.9)	0.665
AWS, yes n (%)	62 (19.4)	4 (7.5)	21 (7.3)	13 (5.3)	<0.001
Pain level median [IQR]	1 [0;2]	2 [1;4]	1 [1;5]	1 [1;4]	<0.001
Anxiety level median [IQR]	3 [2;5]	3 [0;5]	1 [0;5]	0 [0;4]	<0.001

Abbreviations. N, number; ROM, range of motion; AWS, axillary web syndrome; IQR, interquartile range (25th–75th percentile). Note. Continuous variables are summarized as median [IQR] and compared across time windows with the Kruskal–Wallis test; categorical variables are shown as n (%) and compared with Pearson’s χ^2^ test. Column headers report n, and analyses use available cases. Pain and Anxiety were each rated on a 0–10 scale (higher = worse). Statistical significance was set at *p* < 0.05.

**Table 3 cancers-17-03296-t003:** Participant characteristics by postoperative physical therapy utilization.

	Did Not Receive PT N = 820 (51.6%)	Received PT N = 769 (48.4%)	*p* Value
Age, median [IQR]	59 [50;67]	58 [50;67]	0.647
BMI, median [IQR]	25 [22;29]	25 [22;28]	0.165
Surgery type			
Lumpectomy	609 (71.6)	456 (81.6)	0.008
Mastectomy	194 (22.8)	85 (15.2)	
Prophylactic mastectomy	39 (4.6)	11 (2.0)	
Lymph node dissection			
No nodes removed	66 (8.1)	34 (8.8)	0.140
1–4	626 (75.5)	299 (77.3)	
≥5 nodes	126 (15.4)	54 (14.0)	
Breast reconstruction	100 (12.1)	111 (14.7)	0.254
Comorbidity	259 (31.6)	351 (45.6)	<0.001
Chemotherapy	138 (19.4)	162 (21.7)	0.147
Radiation therapy	247 (34.7)	276 (37.0)	0.194
ARM-BCT score, median [IQR]	5 [3;8]	6 [4;8]	<0.001
QuickDASH score, median [IQR]	11 [0;16]	15 [12;18]	<0.001
ROM limitation	141 (36.1)	271 (36.4)	0.478
Decreased function	173 (44.4)	274 (36.8)	0.015
Lymphedema	38 (9.7)	53 (7.1)	0.136
AWS	56 (14.2)	61 (8.2)	0.002
Pain level, median [IQR]	1 [0;4]	1 [1;3]	0.820
Anxiety level, median [IQR]	2 [1;5]	1 [1;3]	<0.001
PT during hospital stay	242 (50.9)	350 (45.5)	0.086
Verbal recommendation from medical staff	41 (8.3)	190 (28.2)	<0.001
Lack of family support	57 (7.4)	69 (9.2)	0.212

Abbreviations. BMI, body mass index (kg/m^2^); PT, physical therapy; ROM, range of motion; AWS, axillary web syndrome; QuickDASH, QuickDisabilities of the Arm, Shoulder and Hand (0–100; higher = greater disability); ARM-BCT, Arm Morbidity following Breast Cancer Treatments (higher = greater risk); IQR, interquartile range. Note. Continuous variables are summarized as median [IQR] (e.g., BMI in kg/m^2^) and compared with Mann–Whitney U; categorical variables are n (% of non-missing) with Pearson’s χ^2^. Column headers shown (non-missing) per PT group; percentages use non-missing denominators. Pain and Anxiety: 0–10 scales (higher = worse). For multi-level variables (e.g., Surgery type, Lymph node dissection), the p value appears on the parent row only.

**Table 4 cancers-17-03296-t004:** Clinical factors associated with early, late, or no initiation of physical therapy after breast cancer surgery.

Factor	No PT vs. Early PT			Late PT vs. Early PT		
	Adjusted OR	95% CI	*p* Value	Adjusted OR	95% CI	*p* Value
Pain (higher level)	1.11	1.01–1.22	0.037	1.21	1.11–1.33	<0.001
Lymphedema	2.59	1.10–6.07	0.029	3.04	1.31–7.04	0.010
ROM limitation	0.80	0.52–1.25	0.323	0.79	0.52–1.20	0.267
AWS	0.87	0.52–1.46	0.587	2.48	1.42–4.34	0.001
Decreased function	1.33	0.84–2.10	0.226	1.00	0.64–1.56	0.995
Anxiety (higher score)	0.98	0.91–1.06	0.623	0.94	0.87–1.01	0.099
Physical activity score (higher)	0.99	0.97–1.02	0.624	1.00	0.98–1.02	0.777
QuickDASH score (higher)	1.00	0.97–1.03	0.774	0.98	0.95–1.01	0.135
ARM-BCT score (higher)	1.08	1.00–1.18	0.062	0.99	0.91–1.07	0.718

Abbreviations. OR, odds ratio; CI, confidence interval; PT, physical therapy; ROM, range of motion; AWS, axillary web syndrome; QuickDASH, Disabilities of the Arm, Shoulder and Hand (0–100; higher = greater disability); ARM-BCT, Arm Morbidity following Breast Cancer Treatments (0–20; higher = greater risk). Note. Multinomial logistic regression of PT timing with three outcome levels: No PT, Early PT (≤3 months), and Late PT (>3 months). Early PT (≤3 months) served as the reference outcome category. Reported values are adjusted odds ratios (OR) with 95% confidence intervals (CI); p values from Wald χ^2^ tests. QuickDASH is reported per 10-point increase; pain/anxiety per 1-point; ARM-BCT per 1-point; categorical predictors are compared with their reference level as specified below. Complete-case sample: n = 1114.

## Data Availability

The minimal de-identified dataset supporting the findings of this study, together with variable metadata (README), is provided in the Appendix A. Additional data are available from the corresponding author upon reasonable request, subject to privacy and ethical restrictions and institutional review board requirements.

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
