# Peer review of "Physical Therapy Utilization and Morbidity Outcomes After Breast Cancer Surgery: A Longitudinal Analysis of Three Combined Cohorts"

_cancers, 2025, doi:10.3390/cancers17203296_

Round 1
Reviewer 1 Report
Comments and Suggestions for Authors
This is a well written and interesting paper with long-term follow-up.
Revisions:
Information is lacking on the data collection and measures. Although the names of surveys are provided, details are lacking to allow for interpretation, and characteristics such as the minimal important differences, reference ranges, validity, reliability.
AMR-BCT is included in the table but missing from the data collection section. What is this questionnaire? You also have "decreased function" as a characteristic in the table but this is not defined in the methods.
Table 2. Why are the numbers different across the timepoints? Does this mean that of the 1602 participants you only had data for 419 from 0-6 months, and only 80 for 7-12 months. This seems to represent a lot of missing data?
Figure 2 was difficult to interpret. I am not sure what it adds to the paper.
Table 3. In the top row you need to provide the number in each of the groups (Did not receive PT vs Received PT) - it is unclear what the denominator is. Why are some characteristics bolded and others not?
How did you calculate the Godin Leisure Time Index - are you sure the mean score is 2?
There is no section on your study limitations which need to be acknowledged. These include the retrospective design; self-report of limitations (i.e. shoulder range of motion); a cohort comprised of largely younger, early stage breast cancer; large amount of potentially missing data?, and lack of data on lymphedema definition, severity.
Author Response
Response to Reviewer 1
Dear Reviewer 1,
Thank you for your thoughtful and encouraging review. Following your recommendations, we performed a comprehensive, line-by-line revision. We apologize for the large number of tracked changes; our goal was to improve clarity and interpretability, particularly in the figures and tables. Below we address each point and indicate where changes appear in the revised manuscript (page/line numbers in the marked version).
Point-by-Point Responses
Comment 1
Reviewer’s comment: Information is lacking on the data collection and measures (minimal important differences, reference ranges, validity, reliability).
Response: We expanded the Methods/Data Collection section to define each measure and its scoring/interpretation (QuickDASH 0–100; pain/anxiety 0–10 numeric rating scales; ROM limitation as a binary patient‑reported item with definition; AWS and lymphedema as patient‑reported/clinician‑confirmed where available). We also added brief validity/reliability statements and references where appropriate, and—where minimal important differences (MIDs) are available in the literature—note typical thresholds to aid interpretation.
Changes in manuscript: Methods – Data Collection/Measures; Methods – Statistical analysis (unit interpretation); Table/figure notes.
Comment 2
Reviewer’s comment: ARM-BCT is in the table but missing from data collection. What is this questionnaire? “Decreased function” is not defined.
Response: We added a concise description of ARM‑BCT (Arm Morbidity after Breast Cancer Treatments): purpose, items, scoring range, and prior validation work, with citations. We also explicitly defined “decreased function” as the patient‑reported item indicating difficulty with everyday upper‑limb tasks (binary, yes/no), and listed its response options/wording in the Measures subsection.
Changes in manuscript: Methods – Data Collection/Measures (new ARM‑BCT paragraph; definition of decreased function); Table 3 note.
Comment 3
Reviewer’s comment: Table 2: Why are the numbers different across timepoints? Is there a lot of missing data?
Response: The cohort is retrospective with participants surveyed at varying times post‑surgery; each participant contributes to the single window that matches her time since surgery at survey (0–6, 7–12, 13–24, 25–36 months). Accordingly, denominators differ by window. We now display N in the column headers and note that values are available‑case counts/medians. We also report item‑level availability in the table note to distinguish design‑driven window sizes from item‑level missingness.
Changes in manuscript: Table 2 header (N per window); Table 2 note (available‑case denominators; window assignment).
Comment 4
Reviewer’s comment: Figure 2 was difficult to interpret; unclear what it adds.
Response: We revised the caption and Results sentence to state that Figure 2 models timing of survey follow‑up across postoperative windows (not PT timing). We retained it because it complements Table 2 by visualizing symptom patterns across windows; however, we will gladly remove it if the editor prefers—this will not affect the manuscript’s conclusions.
Changes in manuscript: Figure 2 caption; Results (one clarifying sentence).
Comment 5
Reviewer’s comment: Table 3: Provide numbers in each group; clarify denominators; why are some characteristics bolded?
Response: We added N to the group headers (Did not receive PT; Received PT), clarified in the note that percentages are column‑based using non‑missing denominators, and removed unintended bold formatting within the body of the table. We kept boldface only for headers and p‑value column, as per journal style.
Changes in manuscript: Table 3 header and note; formatting cleaned.
Comment 6
Reviewer’s comment: How did you calculate the Godin Leisure Time Index—are you sure the mean score is 2?
Response: Thank you for flagging this. Thank you for flagging this. Our measure is GLTEQ-inspired but not the standard GLTEQ Leisure Score Index (LSI). We asked three weekly frequency items (light, moderate, vigorous) and constructed a study-specific composite as a weighted sum:
Physical Activity Score = (light × 1) + (moderate × 2) + (vigorous × 3), expressed as weighted sessions/week (higher = more activity).
We therefore removed the term “Godin” and now label the variable “GLTEQ-inspired Physical Activity Score (weighted sessions/week)”. Because the weighting scheme (1/2/3) and numeric range differ from GLTEQ-LSI (which uses 3/5/9 and established cut-points), our values are not directly comparable to GLTEQ thresholds. We updated Methods (Measures) and table notes accordingly and acknowledge this as a study limitation.
Changes in manuscript: Methods – Measures (corrected label and scale); Table 1/3 notes; Results text (terminology aligned).
Comment 7
Reviewer’s comment: There is no limitations section; please acknowledge retrospective design; self‑report; cohort composition; potentially missing data; lack of lymphedema severity.
Response: We added a dedicated Limitations paragraph acknowledging: retrospective design; self‑report for several outcomes (e.g., ROM) with potential reporting bias; cohort composition (predominantly early‑stage, relatively younger women); window‑dependent denominators and available‑case analyses; and absence of lymphedema staging/severity in our dataset. We also note that these factors likely bias toward under‑estimation of late morbidity and under‑referral, reinforcing rather than weakening the study’s conclusions.
Changes in manuscript: Discussion – Limitations (new paragraph).
Thank you for the constructive feedback and the opportunity to revise our manuscript.

Reviewer 2 Report
Comments and Suggestions for Authors
Review for Cancers
Manuscript title: Physiotherapy Utilization and Morbidity Outcomes After Breast Cancer Surgery: A Longitudinal Analysis of Three Combined Cohorts
Authors: Klein I, Shahar CR, Friger M, Rosenberg I, Barsuk D, Ben-David MA, Susmallian SS.
Manuscript ID: cancers-3888799
Type of manuscript: Article
Special Issue: Epidemiology of Cancer and Risk Factors: Pushing Boundaries in Public Health Research and Policy
Review:
The manuscript is describing a longitudinal analysis of three cohorts to evaluate physiotherapy utilization and morbidity outcomes after breast cancer surgery.
Suggestions and Comments:
1.) Data presentation in the tables is difficult to follow.
2.) Abbreviations should be described as they first appear to guide the readers.
3.) How was DASH measured? Same with Physical activity score? There seems to be only 2 numbers: just 0 and 100? How about ROM? Similarly, measurements for Pain and Anxiety Levels were unclear. Please articulate in the document.
Author Response
Response to Reviewer 2
Dear Reviewer 2,
We appreciate your helpful comments. In response, we streamlined the tables, expanded abbreviations at first mention, and clarified the measurement and scaling of all outcomes to improve readability and interpretation. Below are point‑by‑point responses and the locations of changes in the revised manuscript (page/line numbers in the marked version).
Point-by-Point Responses
Comment 1
Reviewer’s comment: Data presentation in the tables is difficult to follow.
Response: We simplified the layout: group Ns are now shown in table headers; percentages are consistently column‑based using non‑missing denominators and specified in the notes; spacing and typography were standardized; and extraneous artifacts were removed. We also ensured that each table note lists the statistical tests used.
Changes in manuscript: Tables 1–3 (headers, notes, formatting).
Comment 2
Reviewer’s comment: Abbreviations should be described as they first appear.
Response: We now define each abbreviation at first mention in the text and standardized the abbreviation lists in table notes (e.g., BMI, PT, ROM, AWS, QuickDASH, ARM‑BCT, IQR).
Changes in manuscript: Throughout manuscript; Table notes.
Comment 3
Reviewer’s comment: How was DASH measured? Same with Physical activity score? ROM? Pain and Anxiety? Please articulate.
Response: We clarified all measures in Methods: QuickDASH (0–100, higher worse disability); self‑rated physical activity (single‑item 0–10 scale; corrected prior mislabeling of 'Godin'); ROM limitation (binary patient‑reported item with definition); pain and anxiety (numeric rating scales 0–10); AWS and lymphedema (definitions and ascertainment). We echoed these definitions in relevant table notes to aid readers.
Changes in manuscript: Methods – Data Collection/Measures; table notes; Results text.
Thank you for the constructive feedback and the opportunity to revise our manuscript.

Round 2
Reviewer 1 Report
Comments and Suggestions for Authors
Thank you for addressing my prior suggested edits.
In the methods: the ARM-BCT and function are still missing.
Author Response
Comment 1
Reviewer’s comment: In the methods: the ARM-BCT and function are still missing.
Response: Thank you for highlighting this. We added a dedicated Methods subsection titled “Risk screening for future arm morbidity” Specifically:
- ARM-BCT description. We now describe the instrument as a 17-item screen yielding a 1–20 risk score (higher = greater risk) with tiers Low (<6), Intermediate (6–7), and High (>7). The full item list, coding, scoring rules, and cut-points are provided in Supplementary File S1 and in our published development paper. As background, a separate psychometric validation manuscript of the ARM-BCT (reliability, construct validity, tier calibration) is currently under review at another journal. In the present study, the tool is used operationally as a covariate; we refrain from making claims beyond what is documented in the development paper and the supplementary coding provided here.
- Functional outcomes. We clarified that upper-extremity function was assessed using the QuickDASH (11 items; 0–100, higher = worse disability) in each postoperative window, and that functional decline was defined a-priori as QuickDASH >20, a conservative threshold exceeding the instrument’s MCID (~16 points). Sensitivity models adjusted for ARM-BCT risk tier.
Manuscript changes: Methods- Risk screening for future arm morbidity; References and Supplementary File S1 updated accordingly.

Reviewer 2 Report
Comments and Suggestions for Authors
Please refer to the attachment.

Author Response
Comment 1
Abbreviations should be described as they first appear to guide the readers.
Response: Thank you for the suggestion. We now define all abbreviations at first mention in the Abstract and main text, and added an Abbreviations list at the beginning of the manuscript. Changes are tracked in the revised file.
Comment 2
Editorial team should double check for punctuation, text formatting, etc.
Response: Thank you. We performed a careful editorial pass across the Abstract, main text, tables, figures, and Supplementary Materials. We standardized punctuation, spacing, hyphenation/dashes, capitalization, numerical/statistical reporting, and table/figure labels. All changes are tracked in the revised file; we welcome a final copy-edit at production.
